# Training Long-Context, Multi-Turn Software Engineering Agents with Reinforcement Learning

## Abstract

Research on applications of reinforcement learning (RL) to large language models has mostly been focused on single-turn problems, such as mathematical reasoning or single-shot code generation. While these problems can be viewed as token-level multi-turn Markov decision processes (MDPs), this view corresponds to a degenerate case of multi-turn interaction where the environment provides no feedback. This contrasts with many real-world domains, such as software engineering (SWE), which require rich multi-turn interactions with a stateful environment that responds to each action with a non-trivial observation. To bridge this gap, we demonstrate the successful application of RL to this general regime. Our methodology begins with rejection fine-tuning (RFT) using execution feedback to train a policy to follow instructions and formatting effectively, followed by a synchronous RL pipeline using DAPO for iterative improvement. Applying this pipeline to Qwen2.5-72B-Instruct, we increase its Pass@1 on the SWE-bench Verified benchmark from 11% to 39%, substantially improving upon the 20% RFT baseline. On the May and June splits of SWE-rebench, the resulting agent achieves Pass@1 of 35% and 31% respectively, competitive with even larger models such as DeepSeek-V3-0324 or Qwen3-235B-A22B, demonstrating that our methodology offers a practical approach for training capable agents for multi-turn interactive tasks using open-weight models.

## 1 Introduction

Large language models (LLMs) are increasingly being deployed within autonomous agents in complex, real-world domains. Software engineering (SWE) is an especially compelling application area, promising substantial economic impact through automation of debugging, code generation, and software maintenance tasks. However, current approaches to developing effective SWE agents rely predominantly on one of three strategies: (i) combining sophisticated scaffolding with proprietary LLMs (AugmentCode, 2025; Refact AI, 2025; Trae, 2025), (ii) leveraging extensive inference-time scaling techniques (Antoniades et al., 2025), or (iii) supervised fine-tuning (SFT) of open-weight models using demonstrations from stronger teacher models (Yang et al., 2025; Zeng et al., 2025; Pan et al., 2025). While these methods have yielded strong initial results, they are often resource-intensive and depend on powerful, proprietary models. This reality underscores the need for methods that can build comparably effective systems from smaller, open-weight models. Reinforcement learning (RL) offers a promising alternative by directly optimizing an agent's policy through interaction with a responsive environment, potentially achieving stronger performance without reliance on teacher models.

The interactive, structured nature of SWE, where actions produce observable transitions and verifiable outcomes, makes it an ideal domain for RL. Yet, to date, most RL applications for LLMs have been limited to single-turn tasks, such as math reasoning or single-shot code generation, which can be trivially modeled as multi-armed bandits or degenerate MDPs with no intermediate environmental feedback (Figure 1).

In contrast, SWE scenarios require agents to manage stateful, multi-turn interactions. Successfully applying RL in this context involves several key challenges:

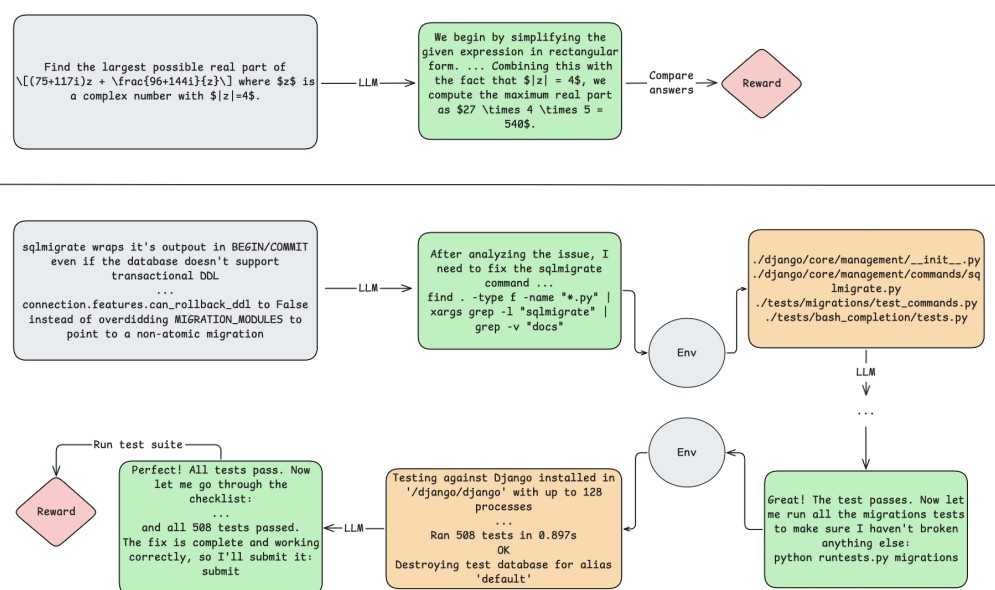

Figure 1: Illustration of task structure differences between bandit-style problems (**top**, e.g., math) and POMDPs (**bottom**, e.g., software engineering), defined in Section 3.1. In bandit settings, the agent takes a single action to produce a final solution based on an initial observation. In contrast, POMDPs require a multi-step interaction loop where the agent repeatedly takes actions and interprets new environmental feedback to guide its subsequent decisions.

- **Long-horizon, multi-turn interaction**: Agents must maintain coherence across dozens of steps with context windows spanning hundreds of thousands of tokens.

- **Complex, informative feedback**: Actions elicit rich outputs (e.g., compiler traces, test logs) that must be interpreted to guide subsequent decisions effectively.

- **Data scalability and fidelity**: Generating high-quality trajectories requires the reproduction of specific repository states in controlled environments, which limits dataset scale. Large-scale datasets such as SWE-SMITH (Yang et al., 2025) and SWE-REBENCH (Badertdinov et al., 2025) begin to address this gap; we primarily build on the latter.

- **Sparse, delayed rewards**: Success signals typically emerge only at the end of long action sequences, complicating credit assignment.

- **Expensive and noisy evaluation**: Unrolling trajectories and subsequent evaluation are costly, and flakiness in tests introduces noise in the reward signal.

In this paper, we address these challenges by developing and validating a complete training pipeline for interactive SWE agents. Our primary contributions are:

- We design and implement a robust two-phase training methodology that combines rejection fine-tuning (RFT) with a subsequent reinforcement learning stage using DAPO (Yu et al., 2025). This pipeline appears to be effective for training agents in long-context interactive environments.

- We provide a strong empirical demonstration of our method's effectiveness. By applying our pipeline to a 72B-parameter open-weight model, we improve its Pass@1 on SWE-BENCH VERIFIED (Chowdhury et al., 2024) from 11% to 39.0%. Coupled with 35% and 31% Pass@1 scores on the SWE-REBENCH May/June splits, the agent achieves competitive performance with powerful, often much larger, open-weight models (Table 1).

- We share key practical insights and analyses from scaling RL to long-context SWE tasks. This includes findings on maintaining training stability and lessons learned that can serve as a blueprint for future work on RL for complex, interactive agents.

## 2 RELATED WORK

**Software engineering agents.** Early systems, notably SWE-agent (Yang et al., 2024), demonstrated that LLMs could effectively operate within sandboxed software environments using predefined toolkits (e.g., shell commands, code editors) and be evaluated via automated unit tests (Jimenez et al., 2024). Subsequent frameworks introduced alternative scaffolding and prompting strategies to enhance model interactions, including Agentless (Xia et al., 2024), OpenHands (Wang et al., 2025), and Moatless (Antoniades et al., 2025). Our work closely aligns with the original SWE-agent design, leveraging similar prompting structures and tool configurations, enabling a direct evaluation of reinforcement learning improvements within this established context.

**Strategies to improve SWE agents.** Prior efforts to enhance SWE agent performance beyond improving scaffoldings broadly focus on either *test-time exploration* or *model-level improvements* through supervised training. Test-time exploration strategies, such as Monte Carlo Tree Search (MCTS) (Antoniades et al., 2025) and guided 1-step lookahead (Zainullina et al., 2025), significantly boost task success by exploring multiple solution trajectories. However, these methods are computationally expensive and sometimes introduce infrastructure complexity due to operations such as environment rollbacks and parallel execution paths.

Alternatively, many efforts have targeted model-level improvements via supervised fine-tuning using expert-curated demonstrations. Prominent examples include SWE-SMITH (Yang et al., 2025), SWE-Fixer (Xie et al., 2025), and Skywork-SWE (Zeng et al., 2025), all of which achieved success by training open-weight models on extensive demonstration data. In contrast, our method relies exclusively on self-generated interaction data obtained through direct RL training. This approach simplifies data collection, eliminates the need for strong teacher models and opens the way for iterative self-improvement.

**Reinforcement learning for coding.** RL has shown notable success in structured reasoning domains like mathematics (Shao et al., 2024; Seed et al., 2025). In the coding domain, early RL applications often focused on single-turn code generation (Dou et al., 2024). In complex SWE tasks, SWE-RL (Wei et al., 2025) is a recent example of applying policy-gradient RL, where the agent achieved strong results. However, its methodology, built upon the Agentless scaffold, frames the problem as a single-turn task where the model generates a complete solution patch in one pass. This approach simplifies the learning problem by sidestepping the complexities of stateful, multi-turn interaction and long-horizon credit assignment.

Our work, alongside other recent advancements, directly addresses these multi-turn challenges. Concurrent to our work, DeepSWE (Luo et al., 2025) successfully scaled critic-free RL training to a 32B-parameter model, while Sky-RL (Cao et al., 2025) introduced an asynchronous RL pipeline for long-context tasks. Our contribution to this emerging area is a complete, multi-stage methodology that successfully applies RL to a larger 72B-parameter model with a 131k token context in a fully interactive, multi-turn setting.

## 3 PRELIMINARIES

### 3.1 TASK FORMULATION

We formalize the task of an autonomous SWE agent as a partially observable Markov decision process (POMDP) (Murphy, 2025), defined by the tuple $\langle \mathcal{Z}, \mathcal{A}, \Omega, T, \mathcal{O}, R, \gamma \rangle$:

- $\mathcal{Z}$: Set of true, latent environment states.
- $\mathcal{A}$: Set of actions available to the agent.
- $\Omega$: Set of observations the agent can receive.
- $T(z'|z, a)$: Transition probability to state $z'$ from state $z$ after action $a$.
- $\mathcal{O}(o|z')$: Probability of receiving observation $o$ from state $z'$.
- $R(z, a)$: Reward provided from taking action $a$ from state $z$.
- $\gamma \in [0, 1]$: Discount factor.

At each step $t$, the environment's latent state $z_t$ is unobservable. Instead, the agent maintains a history $h_t$ of all previous actions and observations: $h_t = (o_0, a_0, o_1, a_1, \ldots, a_{t-1}, o_t)$. A complete history corresponding to a finished episode is called a trajectory $\tau$. The agent's policy, parameterized by $\theta$, selects the next action based on this history: $\pi_\theta(a_t|h_t)$. Due to the LLM's autoregressive nature, the policy probability factorizes over tokens within an action:

$$\pi_\theta(a_t|h_t) = \prod_{k=1}^{|a_t|} \pi_\theta(a_{t,k}|h_t, a_{t,<k}). \tag{1}$$

Here, $a_{t,k}$ is the $k$-th token of the action $a_t$, and $|a_t|$ is the length of the action in tokens.

In our SWE setting, these components are instantiated as follows:

- **Environment State** ($z_t$): Complete, hidden software environment state, including file system, source code, and running processes.
- **Action** ($a_t$): A command string generated autoregressively by the LLM (potentially with reasoning and tool calls).
- **Observation** ($o_t$): The execution output from a command (typically *stdout*, *stderr*, and exit codes). The initial observation $o_0$ contains the natural-language description of the task from the GitHub issue.
- **History** ($h_t$): Complete observable history of past actions and observations, conditioning the policy's next decision.

Our goal is to find a policy $\pi_\theta$ that maximizes the expected cumulative reward. The cumulative reward for a trajectory $\tau$ is the sum of rewards over all its steps, $G(\tau) = \sum_{t=0}^{|\tau|-1} R(z_t, a_t)$, where $|\tau|$ denotes the number of actions in the trajectory. Given our sparse reward structure where immediate rewards $R(z_t, a_t)$ are zero for all non-terminal steps, this simplifies to the terminal reward, which we denote as $R(\tau)$. $R(\tau)$ is 1 if the final proposed patch passes the validation test suite, and 0 otherwise.

## 3.2 FROM PPO TO DAPO

Reinforcement learning has traditionally been dominated by *Proximal Policy Optimization* (PPO) (Schulman et al., 2017), which employs a learned critic to estimate the advantage of each generated action, shown in Equation 5. While powerful, PPO introduces extra overhead and sensitivity to hyperparameter tuning.

*Group-Relative Policy Optimization* (GRPO) (Shao et al., 2024) eliminates the need for a learned advantage estimator. Instead, it computes a Monte Carlo estimate of the advantage for each trajectory by normalizing its terminal reward against the mean and standard deviation of rewards from a group of trajectories sampled from the same policy.

For a given initial observation, GRPO samples a group of $G$ complete trajectories $\{\tau^{(1)}, \ldots, \tau^{(G)}\}$ from the policy and collects their terminal rewards $\{R(\tau^{(1)}), \ldots, R(\tau^{(G)})\}$. The advantage of a trajectory $\tau^{(i)}$ is calculated relative to the average reward of the group, $\bar{R} = \frac{1}{G} \sum_{i=1}^{G} R(\tau^{(i)})$:

$$\hat{A}^{(i)} = \frac{R(\tau^{(i)}) - \bar{R}}{\sigma_R + \delta}, \tag{2}$$

where $\sigma_R$ is the standard deviation of rewards in the group and $\delta$ is a small constant for numerical stability. This single advantage value $\hat{A}^{(i)}$ is then applied to all tokens within the trajectory $\tau^{(i)}$, shown in Equation 7.

DAPO extends GRPO by introducing several practical improvements for enhanced stability and efficiency. Key modifications include:

- **Clip higher**: Instead of a symmetric clip range $(1 - \varepsilon, 1 + \varepsilon)$, DAPO uses asymmetric bounds $(1 - \varepsilon_{low}, 1 + \varepsilon_{high})$. Typically, $\varepsilon_{high}$ is set higher than $\varepsilon_{low}$ to better prevent policy's entropy collapse.

- **Dynamic sampling**: Sampled trajectories that carry no learning signal (i.e., $\hat{A}^{(i)} = 0$) are filtered out, focusing computation on effective updates.

- **Soft overlong punishment**: An incremental penalty is imposed on trajectories that exceed a predefined threshold. The penalty scales linearly within a specified interval and is added to the original test-based reward to discourage excessively long responses.

- **Token-level loss**: The original GRPO algorithm uses sample-level averaging, where each trajectory has equal weight. DAPO suggests averaging the loss over all tokens in the batch, as shown in Equation 8. This method ensures every token across the batch contributes equally to the gradient, giving greater influence to longer trajectories.

Our implementation applies the DAPO framework, adapting the reward function to better suit the multi-turn nature of SWE tasks, as detailed in Section 4.3. Complete mathematical formulations for objective functions are provided in Appendix A.

## 3.3 Agent scaffolding

Our agent follows the SWE-agent (Yang et al., 2024) implementation with a ReAct-style loop (Yao et al., 2023), interacting with the environment through a predefined set of tools. The entire action–observation history conditions every decision. The agent interacts with an environment using the following tools:

- Arbitrary shell commands (*ls*, *cat*, *grep*, etc.).

- An *edit* command that replaces a specified range of lines in a file with new text. The command requires the agent to provide the replacement text with precise indentation and can operate on either the currently open file or one specified by a file path.

- Custom search and navigation utilities (e.g., *search_file*, *open*, *goto*).

- A *submit* command that takes no arguments, signals that the agent has finished its work. This action terminates an episode.

Each SWE task includes a GitHub-style issue with a natural language description, a failing test suite that evaluates final patch correctness, and a sandboxed environment initialized from a repository snapshot. Prompts, available tools and an example of an SWE task can be found in Appendices F-G.

## 4 Training methodology

This section outlines our carefully curated data and a two-phase training pipeline optimized for multi-turn SWE tasks.

## 4.1 Data

We start from the publicly available SWE-REBENCH dataset, containing 21,336 tasks sourced from approximately 3,400 Python GitHub repositories. Each task includes a GitHub issue, a ground-truth solution patch, a validation test suite, and a reproducible environment.

To ensure high-quality and stable training, we apply rigorous filtering criteria, resulting in 7,249 tasks selected according to:

- **Task correctness**: Remove tasks causing test failures due to invalid references or imports (e.g., *AttributeError*, *ImportError*), as indicated in task metadata. These cases would require agent to guess particular identifier names.

- **Controlled complexity**: Include only tasks modifying up to seven files and fewer than 500 lines of code to maintain manageable complexity.

- **LLM-assessed quality**: Exclude tasks with unclear issue descriptions, overly complex tasks, or flawed test patches according to LLM-generated scores from the original dataset. This is done by removing all problems with the assigned LLM score of 3.0 in the metadata field.

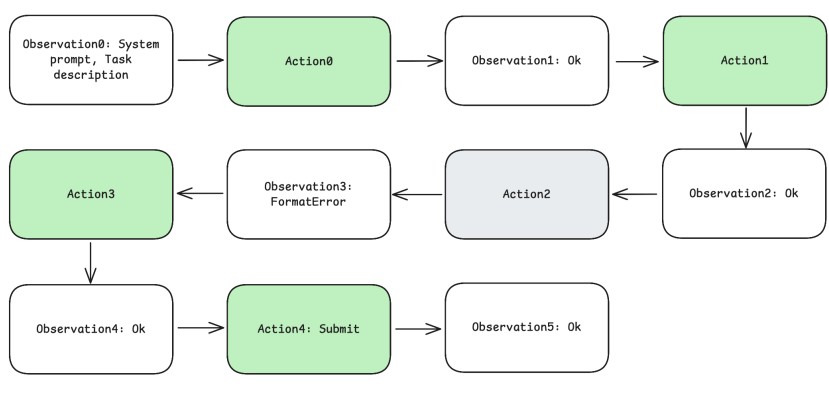

Figure 2: An example trajectory from the agent's interaction used in RFT. Only green (error-free) assistant turns contribute to training loss.

- **Deterministic tests**: Remove tasks with flaky tests that produce inconsistent results across repeated executions (50 trials), ensuring stable training signals. Non-deterministic test behavior occurs mostly due to external service calls or floating-point inaccuracies.

For evaluation, we use the standard SWE-BENCH VERIFIED benchmark, a 50-problem random subset (referred to as VERIFIED-50) for faster intermediate checkpoint evaluation, and the monthly splits of SWE-REBENCH (May-June) which are not included in the training set, ensuring a fair and contamination-free comparison. The full list of VERIFIED-50 problems can be found in Appendix B.

## 4.2 PHASE 1: REJECTION FINE-TUNING (RFT)

We start from the open-weight **Qwen2.5-72B-Instruct** model. Out of the box it achieves only ∼11% Pass@1 on SWE-BENCH VERIFIED with poor instruction following being the dominant issue, resulting in improperly formatted commands. Examples of such behavior can be found in Appendix E.

To address this problem and to warm up the model for the RL stage, we perform rejection fine-tuning. First, we run the initial checkpoint 10 times on the selected SWE-REBENCH tasks and keep only trajectories whose patches pass the test suite. This yields a set of 6,548 successful trajectories, on which we run a single epoch of supervised fine-tuning. During this epoch, we mask assistant turns that triggered an environment-formatting error, thereby focusing the loss only on valid actions and improving adherence to the tool structure (see Figure 2). After RFT, the model's accuracy rises to ∼20% (Table 1) and serves as a baseline for comparison with subsequent RL runs. Hyperparameters for the RFT run are listed in Appendix C.

## 4.3 PHASE 2: MULTI-TURN RL

The core of our work involves applying RL to thousands of problems in an iterative loop. Each RL iteration includes:

- **Problem sampling**: A subset of problems is selected from the training pool.

- **Rollouts generation**: We sample $G = 10$ complete trajectories per problem using the current policy.

- **Reward computation**: Following DAPO formulation, the final reward signal $R_{\text{final}}(\tau)$ combines the binary success reward from test execution with a trajectory length penalty. Unlike the original token-based penalty, we apply a linear penalty for exceeding a predefined number of steps to better reflect the multi-turn nature of SWE tasks. The reward is

computed as follows:

$$R_{\text{final}}(\tau) = R(\tau) + R_{\text{length}}(\tau) \tag{3}$$

$$R_{\text{length}}(\tau) = \begin{cases} 0, & \text{if } |\tau| < L_{\text{thr}} \\ \frac{L_{\text{thr}} - |\tau|}{T_{\text{max}} - L_{\text{thr}}}, & \text{if } |\tau| \geq L_{\text{thr}} \end{cases} \tag{4}$$

Here, $R(\tau) \in \{0, 1\}$ is the terminal reward defined in Section 3.1, $L_{\text{thr}}$ and $T_{\text{max}}$ are hyperparameters representing the penalty threshold and maximum number of turns per trajectory, respectively.

- **Advantage estimation**: Rewards are averaged and normalized within each 10-sample group; samples with zero advantage are dropped.

- **Optimization**: We update all model parameters using DAPO's clipped token-level objective.

Our RL training is divided into two sequential stages. The first stage establishes a baseline policy at a 65k context length. The second stage then advances this policy by training on a longer 131k context and doubling the maximum number of agent turns $T_{\text{max}}$, allowing the agent to tackle more complex problems.

Transitioning to this more computationally demanding setting required adjusting key hyperparameters to ensure stable and efficient training. Following practices from training reasoning models with large-scale RL (Rastogi et al., 2025; Parashar et al., 2025; He et al., 2025), we decreased the high clip bound, increased problem difficulty and batch size, and decreased the number of instances sampled per iteration. The problems difficulty is increased by removing instances that have a success rate of 2/3 over all training iterations. To eliminate instances that are likely unsolvable, we also filter those that have never been solved. The complete set of hyperparameter changes between stages is detailed in Table 3.

This second phase boosts performance to 39.0% on SWE-BENCH VERIFIED. On the held-out SWE-REBENCH evaluation sets, it achieves 35.0% on the May split and 31.7% on the June split. The significant gap between our final Pass@1 score (39.0%) and Pass@10 score (58.4%) suggests that while the agent's single best guess may be incorrect, a valid solution frequently exists within its top proposals. This indicates strong potential for application of re-ranking or best-of-$n$ selection mechanisms to further improve performance.

Table 1: Comparison of our models against open-weight baselines on SWE-BENCH VERIFIED and SWE-REBENCH. Pass@1 metrics are averaged over 10 runs and reported with the standard error of the mean. Our final model and the baseline models are evaluated with a 131k context length; our intermediate models are evaluated at the context length used during their respective training stages (65k). All models use their default decoding parameters as specified in their Hugging Face configuration.

| Model | SWE-bench Verified | | SWE-rebench May | | SWE-rebench June | |
|---|---|---|---|---|---|---|
| | Pass@1 | Pass@10 | Pass@1 | Pass@10 | Pass@1 | Pass@10 |
| Qwen2.5-72B-Inst | $11.4 \pm 0.24$ | 31.0 | $14.5 \pm 1.33$ | 40.0 | $14.6 \pm 1.03$ | 36.6 |
| + RFT @ 65k | $20.5 \pm 0.42$ | 43.0 | $22.5 \pm 1.18$ | 45.0 | $21.0 \pm 1.51$ | 43.9 |
| + Stage 1 RL @ 65k | $35.7 \pm 0.28$ | 54.6 | $36.5 \pm 1.59$ | 55.0 | $31.2 \pm 0.80$ | 53.7 |
| + Stage 2 RL @ 131k | $39.0 \pm 0.50$ | 58.4 | $35.0 \pm 1.54$ | 52.5 | $31.7 \pm 1.31$ | 53.7 |
| Llama-4 Maverick | $15.8 \pm 0.54$ | 47.2 | $19.0 \pm 1.72$ | 50.0 | $13.7 \pm 1.79$ | 39.0 |
| Qwen3-32B no-think | $20.4 \pm 0.34$ | 44.0 | $21.8 \pm 1.54$ | 50.0 | $17.6 \pm 1.30$ | 36.6 |
| gpt-oss-120b | $22.8 \pm 0.54$ | 58.4 | $24.8 \pm 1.26$ | 57.5 | $19.5 \pm 2.36$ | 48.8 |
| Qwen3-235B no-think | $25.8 \pm 0.37$ | 54.4 | $27.3 \pm 1.15$ | 57.5 | $22.9 \pm 2.16$ | 48.8 |
| DeepSeek-V3-0324 | $39.6 \pm 0.47$ | 62.2 | $36.8 \pm 0.92$ | 60.0 | $31.5 \pm 1.38$ | 58.5 |
| Qwen3-235b-Inst-2507 | $46.9 \pm 0.19$ | 69.4 | $41.3 \pm 1.13$ | 57.5 | $38.3 \pm 1.93$ | 61.0 |

## 5 RESULTS AND FINDINGS

### 5.1 MAIN RESULTS

Our two-phase procedure yields substantial improvements. Rejection fine-tuning provides an initial performance boost by enhancing the model's ability to interact correctly with the environment. This is followed by over 100 RL iterations that progressively refine the agent's policy. The performance trends in Figure 3 clearly illustrate how the agent's behavior changes over iterations and stages. In Stage 1, the agent shows consistent improvement before its Pass@1 score begins to plateau. The switch to Stage 2 provides a further performance increase. Notably, this second stage is also characterized by a significant growth in the average steps per trajectory and the number of trajectories finished by a submit command, suggesting that the agent engages in longer, more complex reasoning to solve the more difficult tasks. The final model achieves 39.0% on the full SWE-BENCH VERIFIED.

For head-to-head comparisons, we evaluate **DeepSeek-V3-0324**, **Llama-4 Maverick**, **Qwen3-235B-A22B no-think**, **Qwen3-32B no-think**, **gpt-oss-120b** and **Qwen3-235B-A22B-Instruct-2507** within the same environment and tool setup (see Table 1) on both SWE-BENCH VERIFIED and SWE-REBENCH. To benchmark our final model against specialized SWE agents, we summarize recent results in Table 2.

Table 2: Comparison of specialized multi-turn SWE agents on SWE-BENCH VERIFIED. The "Before" column shows the Pass@1 score of the model prior to training, while the "After" column shows the final performance. "Teacher Distillation" indicates whether a stronger model was used for training.

| Agent | Base Model | Before | After | Teacher Distillation |
|---|---|---|---|---|
| Ours | Qwen2.5-72B-Instruct | 11.4% | 39.0% | No |
| DeepSWE-32B, (Luo et al., 2025) | Qwen3-32B | 23.0% | 42.2% | No |
| SkyRL-Agent-14B-v0, (Cao et al., 2025) | Qwen3-14B | 18.0% | 21.6% | No |
| SWE-Fixer-72B, (Xie et al., 2025) | Qwen2.5-72B-Base | – | 30.2% | Yes |
| SWE-agent-LM-32B, (Yang et al., 2025) | Qwen2.5-Coder-32B-Instruct | 14.3% | 40.2% | Yes |
| Skywork-SWE-32B, (Zeng et al., 2025) | Qwen2.5-Coder-32B-Instruct | 6.4% | 38.0% | Yes |
| SWE-Gym-32B, (Pan et al., 2025) | Qwen2.5-Coder-32B-Instruct | 7.0% | 20.6% | Yes |
| SWESynInfer-72B, (Ma et al., 2024) | Qwen2.5-72B-Instruct | 25.4% | 30.2% | Yes |
| R2EGym-Agent-32B, (Jain et al., 2025) | Qwen2.5-Coder-32B-Instruct | 7.0% | 34.4% | Yes |

### 5.2 FINDINGS

A commonly adopted practice in dataset preparation, also mentioned in DeepSWE (Luo et al., 2025), is to filter or mask trajectories that exceed the model's maximum context length. This is often motivated by the desire to reduce reward noise. However, we find this must be applied with caution. Manually crafted heuristics can introduce biases, breaking the assumption that the training data is sampled from the same distribution as the policy being optimized. In our setting, these long trajectories often occur when the agent is stuck in a repetitive loop. By discarding these trajectories, one also discards specific negative examples of this failure mode. As a result, the agent is not

penalized for this looping behavior and fails to learn how to break out of such cycles, which can lead to it occurring more frequently during training.

We also observe a more subtle instability related to discrepancies between sampling and training. Midway through training, we upgraded the vLLM (Kwon et al., 2023) runtime, which introduced internal changes to decoding parameters. The upgrade enabled *top_k* and *min_p* filtering (previously off by default) using model-dependent values inherited from Hugging Face configurations. While this initially improved evaluation metrics, it caused performance to degrade after 5–10 training iterations. We attribute this to a distribution mismatch that violates the assumptions of importance sampling. The DAPO objective relies on the probability ratio $\rho_{t,k}(\theta)$ to correct for evaluating actions under the current policy $\pi_\theta$ using data generated by the policy from the previous iteration $\pi_{\theta_{old}}$ (see Equation 6). Enabling decoding filters like *top_k* or *min_p* means that trajectories were sampled from a modified, truncated distribution $\pi_{\text{rollout}}$, not the true policy $\pi_{\theta_{old}}$. Consequently, the ratio $\rho_{t,k}(\theta)$ becomes an invalid estimator of the policy change, leading to biased gradient updates and training instability. Once unbiased sampling was restored, performance recovered.

## 6 DISCUSSION AND FUTURE WORK

Our work successfully demonstrates that modern reinforcement learning algorithms, specifically those based on the DAPO framework, can train capable agents for complex, interactive software engineering tasks. This process, however, highlights fundamental challenges in agent-based learning and reveals several key directions for future research:

- **Sparse Rewards and Credit Assignment**: A fundamental challenge is that the agent receives only a single binary success signal at the end of a long trajectory. This sparsity makes it difficult to perform credit assignment, that is, to identify which specific actions in a long sequence were crucial for the outcome. Broadcasting a single advantage estimate across thousands of preceding tokens can result in noisy and inefficient policy updates. Several research directions could address this: (i) reward shaping, which involves designing intermediate rewards based on signals like passing a subset of tests or reducing compiler errors; (ii) training an auxiliary critic or value head to provide step-level advantage estimates, enabling more granular updates; and (iii) prefix sampling, where rollouts are initiated from a shared non-empty trajectory prefix to better isolate the impact of later decisions.

- **Uncertainty and Risk-Awareness**: The binary, success-based reward objective encourages the agent to submit a patch "at any cost", which leads it to act confidently even when a solution is unlikely. For real-world deployment, agents must recognize when to abstain. This requires better uncertainty estimation; for instance, by training the model to explicitly output a confidence score or by using the policy's output entropy as a proxy for uncertainty. Such estimates would enable a precision-recall trade-off, allowing the agent to decide when to halt or to apply more compute for best-of-$n$ selection without an external outcome-supervision model.

## 7 REPRODUCIBILITY STATEMENT

Dataset details, including the initial selection from SWE-REBENCH and the rigorous filtering criteria applied, are described in Section 4.1. The full list of problems used for intermediate evaluation (VERIFIED-50) is provided in Appendix B. Our two-phase training methodology is detailed in Section 4, with the RFT process in Section 4.2 and the two-stage RL process in Section 4.3. All hyperparameters for RFT and both RL stages are fully specified in Appendix C, with key changes highlighted in Table 3. Details of our computational infrastructure, including hardware and software frameworks like vLLM, are provided in Appendix D. Agent's prompts and tools are specified in Appendix F. Finally, the full objective functions for PPO, GRPO, and DAPO as implemented in our work are presented in Appendix A.

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

## A  PPO, GRPO AND DAPO OBJECTIVES

In our setting, the PPO objective can be the following:

$$\mathcal{J}_{\text{PPO}}(\theta) = \mathbb{E}_{\substack{o_0 \sim \mathcal{D}, \\ \tau \sim \pi_{\theta_{old}}(\cdot|o_0)}} \left[ \frac{1}{|\tau|} \sum_{t=0}^{|\tau|-1} \frac{1}{|a_t|} \sum_{k=1}^{|a_t|} \min\Big(\rho_{t,k}(\theta)A_{t,k}, \text{clip}(\rho_{t,k}(\theta), 1-\varepsilon, 1+\varepsilon)A_{t,k}\Big) \right], \quad (5)$$

where $o_0 \sim \mathcal{D}$ indicates that initial observations are sampled from the distribution of tasks in our dataset and $\theta_{old}$ represents parameters of policy from the previous training step.

$$\rho_{t,k}(\theta) \triangleq \frac{\pi_\theta(a_{t,k}|h_t, a_{t,<k})}{\pi_{\theta_{old}}(a_{t,k}|h_t, a_{t,<k})} \quad (6)$$

is the probability ratio for the $k$-th token in action $a_t$, and $A_{t,k}$ is its corresponding advantage estimate.

The GRPO objective function uses the same clipped surrogate objective as PPO but substitutes the critic-based advantage with this group-wise advantage estimate:

$$\mathcal{J}_{\text{GRPO}}(\theta) = \mathbb{E}_{\substack{o_0 \sim \mathcal{D}, \\ \{\tau^{(i)}\}_{i=1}^G \sim \pi_{\theta_{old}}(\cdot|o_0)}} \left[ \frac{1}{G} \sum_{i=1}^G \frac{1}{|\tau^{(i)}|} \sum_{t=0}^{|\tau^{(i)}|-1} \frac{1}{|a_t^{(i)}|} \sum_{k=1}^{|a_t^{(i)}|} \min\Big(\rho_{t,k}^{(i)}(\theta)\hat{A}^{(i)}, \text{clip}(\rho_{t,k}^{(i)}(\theta), 1-\varepsilon, 1+\varepsilon)\hat{A}^{(i)}\Big) \right]. \quad (7)$$

DAPO suggests modifications on top of GRPO described in Section 3.2:

$$\mathcal{J}_{\text{DAPO}}(\theta) = \mathbb{E}_{\substack{o_0 \sim \mathcal{D}, \\ \{\tau^{(i)}\}_{i=1}^G \sim \pi_{\theta_{old}}(\cdot|o_0)}} \left[ \frac{1}{\sum_{i=1}^G \sum_{t=0}^{|\tau^{(i)}|-1} |a_t^{(i)}|} \sum_{i=1}^G \sum_{t=0}^{|\tau^{(i)}|-1} \sum_{k=1}^{|a_t^{(i)}|} \min\Big(\rho_{t,k}^{(i)}(\theta)\hat{A}^{(i)}, \text{clip}(\rho_{t,k}^{(i)}(\theta), 1-\varepsilon_{low}, 1+\varepsilon_{high})\hat{A}^{(i)}\Big) \right]. \quad (8)$$

## B  VERIFIED-50 PROBLEMS

The VERIFIED-50 dataset, which we curate from the SWE-BENCH VERIFIED by randomly selecting problems, contains the following problem instances:

```
sympy__sympy-22080
django__django-15315
django__django-11333
matplotlib__matplotlib-20826
django__django-11532
django__django-16642
django__django-14855
sphinx-doc__sphinx-8721
pylint-dev__pylint-4604
sympy__sympy-13615
django__django-13089
django__django-15987
django__django-14725
sympy__sympy-14248
pytest-dev__pytest-7982
```

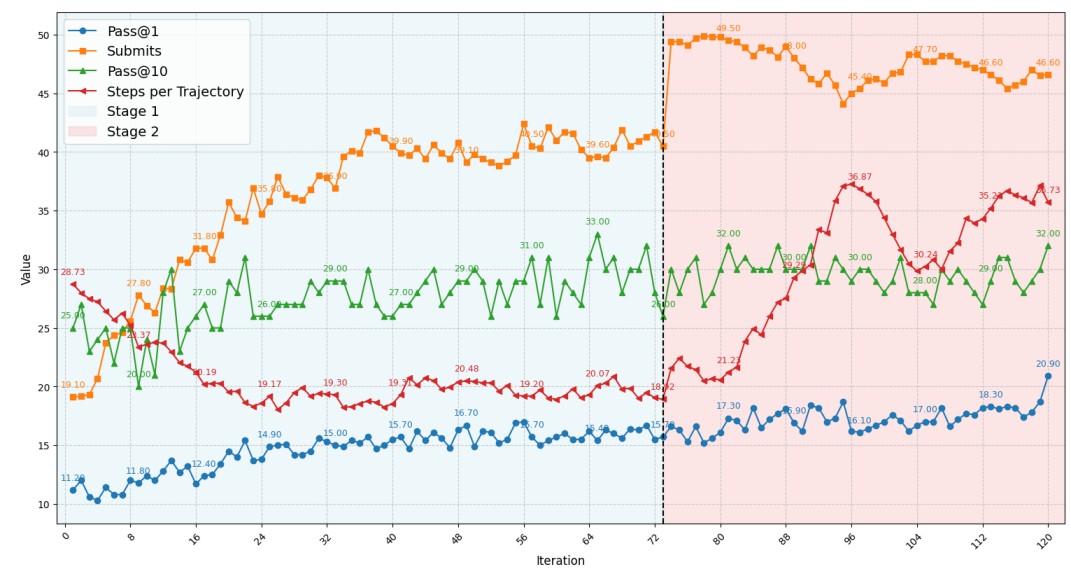

Figure 3: A detailed performance trend of the RL-trained agent over all iterations. Statistics include Pass@1, Pass@10, the number of submit commands and the average number of steps per trajectory. All metrics are computed on VERIFIED-50.

```
django__django-15280
scikit-learn__scikit-learn-13142
pytest-dev__pytest-5809
matplotlib__matplotlib-23299
django__django-16560
django__django-15103
sympy__sympy-16792
django__django-14007
psf__requests-2317
django__django-11880
django__django-16136
django__django-16661
sympy__sympy-17139
sphinx-doc__sphinx-8595
sympy__sympy-14531
django__django-10880
sympy__sympy-19346
sphinx-doc__sphinx-9229
django__django-11265
matplotlib__matplotlib-25332
scikit-learn__scikit-learn-13135
pydata__xarray-6744
pydata__xarray-6461
sympy__sympy-15017
django__django-13417
matplotlib__matplotlib-24870
django__django-15368
django__django-11095
django__django-15554
pydata__xarray-6992
django__django-15863
django__django-13363
sympy__sympy-13852
django__django-14017
pylint-dev__pylint-4661
```

Listing 1: A list of problems from Verified-50.

## C  HYPERPARAMETERS

**Inference.**  For rollout generation, we run the model with a temperature of 1.0, explicitly disabling all other decoding parameters such as *top_p*, *top_k*, *min_p*, *repetition_penalty* and others. This ensures unbiased sampling, which is critical for the validity of importance sampling ratios used during training. We demonstrate the dangers of biased sampling procedures in Section 5.2.

**Training.**  For RFT, we perform one epoch of training at a 65k context length, with a learning rate of $5 \times 10^{-6}$, AdamW optimizer with weight decay of 0.1, 10 warmup steps, and a cosine decay scheduler with *end_lr* $= 0.0$. We use a batch size of 64, resulting in 50 gradient update steps.

For the RL pipeline, we list all hyperparameters changed across stages in Table 3. Both setups share common settings: *gradient_clipping* $= 1.0$; AdamW with $\beta_1 = 0.9$, $\beta_2 = 0.999$, $\varepsilon = 1 \times 10^{-8}$, weight decay of 0.1; learning rate of $10^{-6}$ and *num_epochs* $= 1$.

Table 3: Key hyperparameters across the two RL training stages.

| Hyperparameter | Stage 1 | Stage 2 |
|---|---|---|
| Problems / Iteration | 300 | 100 |
| Total Problems | 7249 | 2028 |
| Batch Size | 128 | 256 |
| $(\varepsilon_{low}, \varepsilon_{high})$ | (0.2, 0.3) | (0.2, 0.26) |
| Maximum Turns ($T_{\max}$) | 40 | 80 |
| Penalty Threshold ($L_{\text{thr}}$) | 10 | 10 |

For the Qwen2.5-72B-Instruct model we use YaRN positional encoding (Peng et al., 2024) with *factor* $= 4.0$ to enable 131k context length training and inference.

## D  INFRASTRUCTURE DETAILS

The described process is based on a fully synchronous RL training process, meaning that inference and training stages are interleaved: once rollout generation is completed, trajectories are used for training. This setup enables fully on-policy training with no policy lag between sampling and updates. However, as described earlier, we sample 10 trajectories for each problem. This results in 2-8 optimization steps per iteration depending on the batch size and the number of trajectories with $\hat{A}^{(i)} \neq 0$.

We believe that asynchronous frameworks can offer greater scalability, but to eliminate complexities like trajectory lag, a synchronous framework is a reasonable choice for these experiments. A key drawback of the synchronous approach is the "straggler" problem: the time for each generation iteration is determined by the single slowest trajectory to complete, which can reduce overall throughput.

To enable full-parameter training on sequences up to 131k tokens, we leverage context parallelism, which partitions long sequences across GPUs. All training and inference are conducted on 16 H200 nodes.

The entire pipeline of distributed agent execution and evaluation is orchestrated at scale using Kubernetes for agent execution and Tracto AI (TractoAI, 2025) for evaluation. During the rollout generation phase, each agent instance runs in a dedicated Kubernetes pod with a resource request of 0.5 CPU and 2 GiB of RAM.

All training and inference are conducted on a cluster of 16 H200 nodes, each equipped with eight GPUs, 32 CPUs and 960 GiB of CPU RAM.

Within this environment, model training is conducted using an internal framework built on JAX, while inference is accelerated using the vLLM framework (Kwon et al., 2023), version 0.7.4.

Figure 4 demonstrates the workflow of a single synchronous iteration. As illustrated, the training process for a new policy begins only after the verification of all trajectories in the batch is complete.

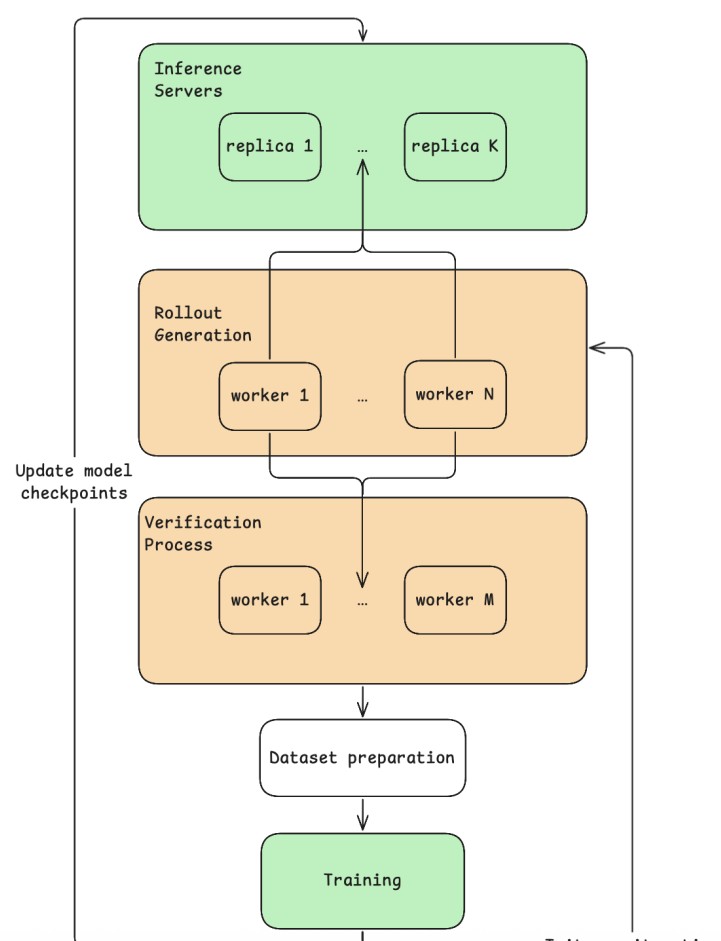

Figure 4: One synchronous iteration of the RL pipeline (green: GPU heavy; yellow: CPU heavy).

## E  EXAMPLE OF AGENT ACTION ERRORS

This section provides examples of common formatting errors produced by the base Qwen2.5-72B-Instruct model prior to rejection fine-tuning. These errors prevent the agent from interacting correctly with the environment and are the primary target of the RFT phase.

**Error Type 1: Command Not Extractable.**  In this common failure mode, the agent correctly reasons about the next step but fails to format its intended action as a distinct, parsable command beginning with ```command and ending with ```. The scaffolding cannot extract a valid command, resulting in an error.

```
Now that we have located the `__deepcopy__` method in `fields.py`, let's
    modify it to deeply copy the `error_messages` dictionary. This will
    ensure that each instance of the form field has its own independent
    copy of the error messages.

### Step 2: Modify the `__deepcopy__` method
I will add a line to the `__deepcopy__` method to deep copy the `
    error_messages` dictionary.

```edit 198:198
    result.error_messages = copy.deepcopy(self.error_messages, memo)
```
```

Listing 2: An example of the 'cannot_extract_command' error type.

**Error Type 2: Multiple Commands in a Single Turn.** The agent is designed to follow a strict ReAct loop, where it emits a single thought and a single action per turn before waiting for an observation. In this error case, the agent violates this protocol by attempting to issue multiple distinct commands within a single turn, leading to a parsing failure.

```
To resolve this issue, I will follow these steps:

1. **Reproduce the Error**: I need to create a Python script that
    reproduces the `KeyError` when passing a `CCDData` instance into `np.
    min()`.
2. **Identify the Cause**: Once the error is reproduced, I will
    investigate the code in `astropy/nddata/mixins/ndarithmetic.py` to
    understand why the `KeyError` is being raised.
3. **Fix the Issue**: I will modify the code to handle the `KeyError`
    appropriately and ensure that `np.min()` works as expected.
4. **Verify the Fix**: I will run the same script to confirm that the `
    KeyError` is no longer raised and that the correct minimum value is
    returned.
5. **Run Existing Tests**: I will run the existing tests in the
    repository to ensure that my changes do not break any other
    functionality.

Let's start by creating a Python script to reproduce the error.

```command
create reproduce_error.py
```
```command
edit 1:1 << 'EOF'
import numpy as np
from astropy.nddata import CCDData

a = CCDData([1, 2, 3], unit="adu")
print(np.min(a))
EOF
```

Now, I will run the script to reproduce the error.

```command
python reproduce_error.py
```
```

Listing 3: An example of the 'multiple_commands_extracted' error type.

## F    AGENT PROMPTS AND AVAILABLE TOOLS

This section details the prompts used to guide the agent. The agent's behavior is primarily governed by a system prompt that defines the required ReAct-style interaction format, and a description of the available tools. This system prompt is provided to the agent at the beginning of every episode for all experiments.

For the initial data collection phase using the base instruct-tuned checkpoint, we appended a one-shot demonstration to the system prompt. This in-context example of a successful trajectory was used solely to improve the quality of the initial rollouts for building the RFT dataset. Crucially, this demonstration was not included in the prompt for the RFT and subsequent RL-trained models; they were trained and evaluated using only the system prompt detailed below.

## F.1 SYSTEM PROMPT

```
You are an autonomous programming agent. Your goal is to resolve the
    issue given to you.
You are given access to a terminal environment with some special tools
    to make your job easier.
You must use the terminal to gain information about the codebase, find
    or modify the relevant files in order to resolve the issue.
In this environment, all standard unix commands (e.g. grep, sed, echo
    etc.) will be available to you.
However, the environment does NOT support interactive session commands
    that expect user input (e.g. vim), so please do not invoke them, it
     will result in an error.
You can however create python scripts and run them, this is very useful
     to reproduce errors or test something.
If some packages are missing, you can install them using an appropriate
     package manager (e.g. pip, apt, etc.).
Do not ask any questions to the environment, it's an automated system
    that can only execute your commands.
When you are satisfied with the changes you made, you should explicitly
     submit them using a special command. This will terminate your
    session.

# SPECIAL TOOLS

In addition to standard unix commands you can use special tools
    described below.
Please note that some of these commands work with the currently open
    file, so pay attention to what file is open.

Usage: create [OPTIONS] FILENAME
  Creates and opens a new filename with the given name.

Usage: edit [OPTIONS] LINE_RANGE [REPLACEMENT_TEXT]
  Replaces lines in LINE_RANGE=<start_line>:<end_line> (inclusive) with
       the
  given text in the currently open or specified file. The
      REPLACEMENT_TEXT
  will be used as provided including all whitespaces, so make sure your
  indentation is correct.
  To input multiple lines into REPLACEMENT_TEXT, you may use the
      following
  syntax:
  ```
  edit 1:1 << 'EOF'
  Line1
  Line2
  EOF
  ```
  You can also provide the file to edit via `--file` option.
  ```
  edit --file path/to/file 1:1 "Your Replacement Text Here"
  ```
  Please note that THIS COMMAND REQUIRES PROPER INDENTATION. If you'd
      like to
  add the line '        print(x)' you must fully write that out, with
      all
  those spaces before the print statement!
Options:
  --file PATH  The file to edit. (If not provided, edits the currently
      open
                  file)

Usage: goto [OPTIONS] LINE_NUMBER
```

```
   Navigates the current window to a given line in the currently open
       file.

Usage: open [OPTIONS] [FILE] [LINE_NUMBER]
  Opens the file at the given path in the editor. If file is not
      specified,
  the last open file will be reopened. If line_number is provided, the
      current
  window will move to show that line.

Usage: replace [OPTIONS] SEARCH REPLACE
  Replaces a given string with another string in the currently open
      file.
Options:
  --replace-all  Replace all occurrences of the SEARCH text.

Usage: scroll_down [OPTIONS]
  Scroll down the window in the currently open file and output its
      contents.

Usage: scroll_up [OPTIONS]
  Scroll up the window in the currently open file and output its
      contents.

Usage: search_file [OPTIONS] SEARCH_TERM [FILE]
  Searches for SEARCH_TERM in file. If FILE is not provided, searches
      in the currently open file.

Usage: submit [OPTIONS]
  Submits your current code and terminates the session.

# ENVIRONMENT RESPONSE

At the very beginning the environment will provide you with an issue
    description. In response to every command that you invoke,
the environment will give you the output of the command or an error
    message followed by a shell prompt.
The shell prompt will be formatted as follows:
```
(Current directory: <current_dir>, current file: <current_file>) bash-$
```
so that you always know what the current directory is and what file is
    currently open.

# YOUR RESPONSE

Your response should consist of two parts: reasoning (arbitrary text)
    and command (surrounded by triple ticks and a special 'command'
    keyword).
Your response should always include A SINGLE reasoning and A SINGLE
    command as in the following examples:

<response example>
First I'll start by using ls to see what files are in the current
    directory. I'll look at all files including hidden ones.
```command
ls -a
```
</response example>

<response example>
Let's search the file `models.py` for the UserEntity class definition.
```command
search_file "class UserEntity" models.py
```

```
```
</response example>

Everything you include in the reasoning will be made available to you
    when generating further commands.
If you'd like to issue two command blocks in a single response, PLEASE
    DO NOT DO THAT! THIS WILL RESULT IN AN ERROR.

# HANDLING TESTS

* You can run existing tests to validate the changes you made or make
    sure you didn't break anything.
* If missing packages or some environment misconfiguration is
    preventing you from running the tests, you can install missing
    packages or fix the environment.
* However UNDER NO CIRCUMSTANCES should you modify existing tests or
    add new tests to the repository.
  This will lead to an error in the system that evaluates your
      performance. Instead, you can just create a temporary script, use
       it to test changes and remove it before submitting.
* If existing tests break because they need to be updated to reflect
    the changes you made, just ignore it. Evaluation system will not
    take it into account.
* However if existing tests are broken because your fix is incorrect,
    you should fix your code and make sure all tests pass before
    submitting the change.

# USEFUL ADVICE

* As a first step, it might be a good idea to explore the repository to
     familiarize yourself with its structure.
* You should also come up with a rough plan of how to resolve the issue
     and put it into your reasoning.
* If the issue description reports some error, create a script to
    reproduce the error and run it to confirm the error. THIS IS
    USUALLY A VERY GOOD FIRST STEP!
* Edit the source code of the repo to resolve the issue
* Rerun your reproduce script and confirm that the error is fixed! THIS
     IS QUITE IMPORTANT!
* Think about edge cases and make sure your fix handles them as well.
* Make sure your solution is general enough and not hardcoded to the
    specific cases reported in the issue description.
* It might be a good idea to ensure that existing tests in the
    repository pass before submitting the change. Otherwise it's easy
    to break existing functionality.
```

Listing 4: System prompt defining the agent's task and tools.

## F.2 ONE-SHOT DEMONSTRATION

The following demonstration was provided as a one-shot example to the base instruct model to generate the initial dataset for RFT.

```
# DEMONSTRATION

Here is a very simple demonstration of how agent can interact with the
    environment using the provided interface.

<demonstration><environment>
# ISSUE DESCRIPTION

Here is a script that is supposed to print out first 10 prime numbers,
    but it seems to have a bug. Can you fix it?
```

```
```
def is_prime(n):
    if n <= 1:
        return False
    for i in range(2, int(n**0.5)):
        if n % i == 0:
            return False
    return True

primes = []
num = 2
while len(primes) < 10:
    if is_prime(num):
        primes.append(num)
    num += 1

print("First 10 prime numbers:", primes)
```

# ADDITIONAL ADVICE

Since you are given a git repository, you can use git commands to
    simplify your work.
For example, if you made a mistake and want to revert your changes, you
     can use `git checkout <file>` to restore the file to its original
     state.
You can also reset all changes in the repository using `git reset --
    hard` command.
Additionally, you can use `git stash` and `git stash pop` to
    temporarily save your changes and restore them later.
Finally, you can see the changes that you've made (which will be
    submitted when you call `submit`) using `git status` or `git diff`
    commands.
However you don't need to use `git add` or `git commit` before
    submitting your changes. In fact,
`submit` only submits currently unstaged changes because it uses `git
    diff` to get the changes that need to be submitted.

# CHECKLIST

Before submitting your solution, please go over the following checklist
     and make sure you've done everything:
- [ ] If an error was reported in the issue description, I have
    successfully reproduced it.
- [ ] If an error was reported in the issue description, I have
    confirmed that my fix resolves the error.
- [ ] I have thought about edge cases and made sure my fix handles them
     as well.
- [ ] I have run existing tests in the repository that might have been
    affected by the change I made and confirmed that they pass.
I want you to list every bullet from this checklist and write your
    reasoning for why you think you did it or didn't need to.

Repository has been uploaded and your shell is currently at the
    repository root. Time to solve the issue!

(Current directory: /project, current file: none) bash-$
</environment>
<agent>
Let me analyze the issue and fix the prime number generator script step
    by step.

First, let's create a Python script to reproduce the reported issue and
     verify the current behavior.
```

```
```command
create prime_numbers.py
```
</agent>
<environment>
[File: prime_numbers.py (1 lines total)]
1:

(Current directory: /project, current file: none) bash-$
</environment>
<agent>
Now I'll add the provided code to the file to test it and see what
    output it produces. The issue description mentions there's a bug,
    but doesn't specify what exactly is wrong. First we need to
    identify the issue.

```command
edit 1:1 << 'EOF'
def is_prime(n):
    if n <= 1:
        return False
    for i in range(2, int(n**0.5)):
        if n % i == 0:
            return False
    return True

primes = []
num = 2
while len(primes) < 10:
    if is_prime(num):
        primes.append(num)
    num += 1

print("First 10 prime numbers:", primes)
EOF
```
</agent>
<environment>
File updated. Please review the changes and make sure they are correct
    (correct indentation, no duplicate lines, etc). Edit the file again
     if necessary.
[File: /project/prime_numbers.py (16 lines total)]
1:def is_prime(n):
2:    if n <= 1:
3:        return False
4:    for i in range(2, int(n**0.5)):
5:        if n % i == 0:
6:            return False
7:    return True
8:
9:primes = []
10:num = 2
11:while len(primes) < 10:
12:    if is_prime(num):
13:        primes.append(num)
14:    num += 1
15:
16:print("First 10 prime numbers:", primes)

(Current directory: /project, current file: none) bash-$
</environment>
<agent>
Now let's run the script to see what output it produces and identify
    the bug.
```

```
```command
python prime_numbers.py
```
</agent>
<environment>
First 10 prime numbers: [2, 3, 4, 5, 6, 7, 8, 9, 11, 13]

(Current directory: /project, current file: none) bash-$
</environment>
<agent>
Ah, I see the issue! The script incorrectly includes numbers 4, 6, 8,
    and 9 in the output as "prime" numbers when they're actually not
    prime. The bug is in the `is_prime()` function - the range should
    go up to `int(n**0.5) + 1` because the upper bound in Python's
    range is exclusive. Let's fix this by modifying the range in the `
    is_prime()` function.

```command
edit 4:4 << 'EOF'
    for i in range(2, int(n**0.5) + 1):
EOF
```
</agent>
<environment>
File updated. Please review the changes and make sure they are correct
    (correct indentation, no duplicate lines, etc). Edit the file again
     if necessary.
[File: /project/prime_numbers.py (16 lines total)]
1:def is_prime(n):
2:    if n <= 1:
3:        return False
4:    for i in range(2, int(n**0.5) + 1):
5:        if n % i == 0:
6:            return False
7:    return True
8:
9:primes = []
10:num = 2
11:while len(primes) < 10:
12:    if is_prime(num):
13:        primes.append(num)
14:    num += 1
15:
16:print("First 10 prime numbers:", primes)

(Current directory: /project, current file: none) bash-$
</environment>
<agent>
Now let's run the script again to verify that the fix works and
    produces the correct list of prime numbers.

```command
python prime_numbers.py
```
</agent>
<environment>
First 10 prime numbers: [2, 3, 5, 7, 11, 13, 17, 19, 23, 29]

(Current directory: /project, current file: none) bash-$
</environment>
<agent>
The output now correctly shows the first 10 prime numbers. Let's go
    through the checklist:
```

```
1242  1. Reproduced the error: Yes, we saw the script incorrectly included
1243     non-prime numbers in its output.
1244  2. Confirmed the fix resolves the error: After modifying the range, the
1245     script now correctly lists only prime numbers.
1246  3. Edge cases considered: The fix handles all cases correctly - numbers
1247     <=1 are non-prime, and the +1 ensures we check all potential
1248     divisors up to the square root.
1249  4. Existing tests: Since this is a standalone script and there are no
1250     existing tests mentioned in the repository, this point isn't
1251     applicable.

      The fix is complete and correct. We can now submit the changes.

      ```command
      submit
      ```
      </agent>
      </demonstration>
```

Listing 5: Demonstration prompt used for initial data collection only.

# G  EXAMPLE OF AN SWE TASK

This section provides an example of a single software engineering task in the raw JSON format used during our training and evaluation process. Each task contains several key fields. The `problem_statement` field holds the natural language bug report from a GitHub issue, which is the initial observation provided to the agent. The agent's goal is to produce a patch that resolves this issue. Task correctness is verified using the `test_patch`, which typically introduces a new unit test that fails on the buggy code but passes once a correct solution is applied. The ground-truth solution is also included as the `patch` field for reference. This task structure follows the format used in the SWE-bench benchmark and is provided here for convenience.

```
{
    "repo": "deepchem/deepchem",
    "instance_id": "deepchem__deepchem-2802",
    "base_commit": "9ef6c58eefd4e5f9bee40743ca14defa6f764f80",
    "patch": "diff --git a/deepchem/data/datasets.py b/deepchem/data/
        datasets.py
    index f80a399ca..dd5ba637f 100644
    --- a/deepchem/data/datasets.py
    +++ b/deepchem/data/datasets.py
    @@ -2286,6 +2286,7 @@ class DiskDataset(Dataset):
    basename = "shard-%d" % shard_num
    DiskDataset.write_data_to_disk(self.data_dir, basename, X, y, w, ids)
    self._cached_shards = None
    +    self.legacy_metadata = True

    def select(self,
        indices: Sequence[int],",
    "test_patch": "diff --git a/deepchem/data/tests/test_setshard.py b/
        deepchem/data/tests/test_setshard.py
    new file mode 100644\nindex 000000000..0fcf4b03e
    --- /dev/null
    +++ b/deepchem/data/tests/test_setshard.py
    @@ -0,0 +1,21 @@
    +import deepchem as dc
    +import numpy as np
    +
    +
    +def test_setshard_with_X_y():
    +  """Test setsharding on a simple example"""
    +  X = np.random.rand(10, 3)
    +  y = np.random.rand(10,)
```

```
+   dataset = dc.data.DiskDataset.from_numpy(X, y)
+   X_shape, y_shape, _, _ = dataset.get_shape()
+   assert X_shape[0] == 10
+   assert y_shape[0] == 10
+   for i, (X, y, w, ids) in enumerate(dataset.itershards()):
+       X = X[1:]
+       y = y[1:]
+       w = w[1:]
+       ids = ids[1:]
+       dataset.set_shard(i, X, y, w, ids)
+   X_shape, y_shape, _, _ = dataset.get_shape()
+   assert X_shape[0] == 9
+   assert y_shape[0] == 9",
"problem_statement": "Bug in dataset.get_shape() when used after
     dataset.set_shard()
## \ud83d\udc1b Bug

## To Reproduce

<!-- If you have a code sample, error messages, stack traces, please
     provide it here as well. -->

## Expected behavior

```
import deepchem as dc
import numpy as np
X = np.random.randn(10, 3)
y = np.random.randn(10)

dataset = dc.data.DiskDataset.from_numpy(X, y)
dataset.get_shape()  # Output: ((10, 3), (10,), (10,), (10,))

for i, (X, y, w, ids) in enumerate(dataset.itershards()):
    X = X[1:]
    y = y[1:]
    w = w[1:]
    ids = ids[1:]
    dataset.set_shard(i, X, y, w, ids)

dataset.get_shape()
# Output: ((10, 3), (10,), (10,), (10,))
# Expected output: ((9, 3), (9, ), (9, ), (9,))
```

Edit 1:

This prints correctly:
```
for i, (X, y, w, ids) in enumerate(dataset.itershards()):
    print (X.shape)  # Output: (9, 3)
```
## Environment
* DeepChem version: 2.6.0.dev

## Additional context
The fix is probably a simple one.",
"hints_text": "",
"created_at": "2022-01-01T16:38:29+00:00",
"version": 0.0,
"FAIL_TO_PASS": [
    "deepchem/data/tests/test_setshard.py::test_setshard_with_X_y"
],
"PASS_TO_PASS": [],
```

```
    "environment_setup_commit": "2401580b6f41fe72f1360493ee46e8a842bd04ba
        ",
    "meta": {
        "failed_lite_validators": [],
        "has_test_patch": true,
        "is_lite": true
    },
    "pull_number": 2802.0,
    "issue_numbers": [
        2772
    ]
}
```

Listing 6: Example of an SWE task ('deepchem__deepchem-2802') in JSON format.

## H  THE USE OF LARGE LANGUAGE MODELS

During the preparation of this manuscript, we utilized a large language model as a writing and editing assistant. The role of the LLM was strictly limited to improving the clarity, grammar, and readability of the text. This included tasks such as rephrasing sentences for better flow, correcting spelling and punctuation, and ensuring consistent style. The LLM was not used for any part of the core research process, including ideation, experimental design, data analysis, or the formulation of conclusions.

