# OpenReview forum: "Training Long-Context, Multi-Turn Software Engineering Agents with Reinforcement Learning"
_ICLR.cc/2026/Conference — Submitted to ICLR 2026_

### Official Review · Reviewer_18P6 · 2025-10-26

**Soundness:** 3
**Presentation:** 2
**Contribution:** 3
**Rating:** 4
**Confidence:** 3

**Summary:**

This paper presents a comprehensive RL pipeline for training long-context, multi-turn software engineering agents. The authors employ a two-stage training process: (1) rejection fine-tuning (RFT) for instruction and formatting alignment, and (2) DAPO-based reinforcement learning for iterative improvement. The proposed agent significantly outperforms the baseline and competitive with larger models.

**Strengths:**

1. The paper demonstrates that RL can substantially improve the capability of SWE agents in multi-turn, interactive settings.
2. The integration of RFT and DAPO is practical and carefully tuned for stability under long context lengths.
3. Results on multiple benchmarks and comparisons prove good results.

**Weaknesses:**

1. The method mostly applies existing components (RFT + DAPO) to a new domain. It brings the potential risk of lacking the conceptual or algorithmic innovation beyond careful system design.
2. The empirical study could be strengthened by isolating the effects of individual components proposed in this work.
3. The authors treat SWE-ReBench as an interactive environment, the model is repeatedly trained within the same repository. This setup may lead to environment overfitting, where the agent learns to exploit the reward dynamics of SWE-ReBench rather than genuinely generalizing to unseen repositories. The authors are recommended to evaluate on unseen environments or cross-repository splits to validate generalization.

**Questions:**

Please refer to the weakness section. My main concern is the incomplete ablations and generalization studies.

---

> ### Author Response · Authors · 2025-11-20
> **Authors response**
>
> We thank the reviewer for the thoughtful feedback. We address the reviewer's points below:
>
> 1: While our method builds on existing components like DAPO, our primary contribution is in the methodology of adapting and scaling these tools to a domain they were not tested for: long-context, multi-turn, stochastic environments. Our work addresses the novel challenge of how to make RL stable and effective in this complex regime. A vanilla application is insufficient; as we note, other work [1] applying RL to a Qwen3-14B model saw only a minor gain (18% to 21.6%). Our novelty lies in the full pipeline, including the RFT warm-up, the two-stage RL curriculum, and the long-context stabilization, that demonstrates how to successfully train a 72B parameter agent for this task.
>
> 2: We agree that isolating component effects is crucial. We provided an initial ablation in Table 1, which demonstrates the distinct, cumulative gains from the RFT phase, the Stage 1 RL, and the Stage 2 RL. To further strengthen the paper, we will include following information in the final version of the paper:
>
> - Performance of RL runs with biased/unbiased sampling, mentioned in Section 5.2 (performance degradation from 35% to 18.8% within 28 training steps).
> - RL Stage 1 and Stage 2 evaluations at the context of 65k with 40 turns and 131k with 80 turns. This directly isolates the impact of our Stage 2 training protocol from the simple increase in context length. The results are as follows:
>
> Stage 1 evaluated @ 65k 40turns: 35.7 ± 0.3%, pass@10: 54.6%
>
> Stage 1 evaluated @ 131k 80turns: 35.4 ± 0.3%, pass@10: 54.0%
>
> Stage 2 evaluated @ 65k 40turns: 37.9 ± 0.3%, pass@10: 56.4%
>
> Stage 2 evaluated @ 131k 80turns: 39.0 ± 0.5%, pass@10: 58.4%
>
> 3: We respectfully clarify a potential misunderstanding regarding the SWE-rebench dataset. The reviewer correctly notes that overfitting to a single environment is a risk. However, our training set is not a single environment; it is composed of 7,249 distinct tasks from 2,022 different GitHub repositories. Most importantly, we rigorously ensured zero intersection with the evaluation benchmarks. For SWE-bench Verified, we ensured no repository-level overlap. Therefore, our agent is evaluated exclusively on tasks it has never seen during training.
>
> [1] Shiyi Cao et al. SkyRL-v0: Train Real-World Long-Horizon Agents via Reinforcement Learning

---

### Official Review · Reviewer_qpff · 2025-10-29

**Soundness:** 3
**Presentation:** 1
**Contribution:** 2
**Rating:** 2
**Confidence:** 4

**Summary:**

Authors introduce a post-training pipeline for training a software engineering (SWE) agent. They construct the training dataset by carefully filtering SWE-Rebench dataset. Then, authors employ two-stage training. The first stage is Rejection Fine-tuning: authors generate 10 trajectories per prompt, and SFT on successful trajectories. In the second stage, DAPO is employed with the outcome feedback; length penalty is added to the reward. These methods make a significant improvement on the well-established SWE-Bench Verified benchmark.

**Strengths:**

Quality: The proposed method can serve as a useful, clean baseline for future research on SWE tasks. Unlike many of the previous papers, authors make minimal dependency on proprietary LLMs for data generation, therefore there is neither dependency the capability of these LLMs nor  terms-of-service issues with proprietary LLMs. Also, authors employ well-established techniques: two-stage post training pipeline with SFT and RL, DAPO algorithm for RL, length penalty on the reward. Adoption of standard techniques provide bigger confidence on the reliability of the findings. Also, this makes it easier for the proposed method to adopt subsequent methodological improvements. Therefore, the quality of the execution is notable.

Significance: As mentioned above, the paper could establish a clean baseline for the task, which will be significant. The performance improvement, however, is not significant as comparable results can be achieved with better scaffoldings.

Originality: RL on SWE tasks is well-established. Multi-turn RL might've been absent at the time of submission, but there are concurrent work such as CWM https://ai.meta.com/research/publications/cwm-an-open-weights-llm-for-research-on-code-generation-with-world-models/ . In any case, I don't think ICLR readers would be surprised that RL on multi-turn SWE task works, as RL on multi-turn agentic tasks are already common.

Clarity: The paper is straightforward to read and understand.

**Weaknesses:**

The work is not presented in a way that highlights authors' main contributions. First five pages are devoted to literature review and problem formulation. While I appreciate authors' attempt to make the paper self-contained and rigorous, this method of presentation is at the cost of not leaving much room for authors to discuss their key contributions. For example, I find PPO->GRPO->DAPO discussion to be quite tangential to authors' key contributions; authors could simply say they use DAPO and move on.

From my perspective, key technical innovations of this paper is majorly 1) how SWE-Rebench data were carefully processed, and 2) how various hyperparameters of RL were set for strong results, given authors use standard methodology for the rest (DAPO, 2-stage training, etc). Unfortunately, these contributions are briefly described without detailed analysis, discussion, and ablation.

**Questions:**

Line 269: which LLM was used for the quality estimation, and what was the prompt?

---

> ### Author Response · Authors · 2025-11-20
> **Authors response**
>
> We thank the reviewer for their thoughtful feedback and for recognizing our work as a "high-quality execution" and a "clean, useful baseline" for the field. We address the reviewer's points below:
>
> Our work fills a gap in understanding how RL scales to large models (>8B) and long contexts (>8k) in multi-turn scenarios. A vanilla application of RL often yields modest gains (e.g., SkyRL: 18.0% -> 21.6% with Qwen3-14B, [1]). We note that works like DeepSWE and CWM are concurrent or released post-submission, validating the importance of this direction. We also believe that at least brief preliminaries are needed for readers unfamiliar with DAPO/POMDP formulation.
>
> During rebuttal, we generated the following data:
>
> 1: We demonstrated how biased versus unbiased sampling influences the performance of the RL model. A run using biased sampling degraded from 35% to 18.8% within just 28 training steps.
>
> 2: We evaluated Stage 1 and Stage 2 checkpoints with contexts of 65k with 40 turns and 131k tokens with 80 turns to isolate the effect of the second stage training from simply extending the context length. The results on the full SWE-bench Verified, averaged across 10 runs, are as follows:
>
> Stage 1 evaluated @ 65k 40turns: 35.7 ± 0.3%, pass@10: 54.6%
>
> Stage 1 evaluated @ 131k 80turns: 35.4 ± 0.3%, pass@10: 54.0%
>
> Stage 2 evaluated @ 65k 40turns: 37.9 ± 0.3%, pass@10: 56.4%
>
> Stage 2 evaluated @ 131k 80turns: 39.0 ± 0.5%, pass@10: 58.4%
>
> All new ablations will be included in the final version of the paper.
>
> **Questions:**
>
> 1: Regarding Line 269, we utilize the SWE-rebench dataset, which already includes annotated scores from an LLM. According to the original paper, the authors fine-tuned Qwen2.5-72B-Instruct to predict SWE-bench Verified labels for Test Patch Correctness, Task Complexity, and Issue Clarity. Further details can be found in Appendix F of that work.
>
> [1] Shiyi Cao et al. SkyRL-v0: Train Real-World Long-Horizon Agents via Reinforcement Learning

---

### Official Review · Reviewer_EQbx · 2025-10-30

**Soundness:** 2
**Presentation:** 2
**Contribution:** 2
**Rating:** 4
**Confidence:** 3

**Summary:**

The paper deals with a common limitation in current reinforcement-learning (RL) application for large language models (LLMs); that is, most prior works focus on single-turn tasks (e.g., math or single-shot code generation), ignoring the multi-turn nature of tasks such as software engineering.
In this paper, a two-phase training pipeline combining rejection fine-tuning (RFT) with following RL via dynamic sampling policy optimization (DAPO) is proposed, and the effectiveness of this pipeline is demonstrated by empirical results on Qwen2.5-72B-Instruct, which supports long-context use case.

**Strengths:**

1. The research topic is clear, practical and timeliness.

The research gap that most LLM-with-RL studies focus on single-turn tasks, while many real-world tasks like SWE require long-horizon and interactive reasoning. To address this gap, this paper provides a reasonable design for the practical SWE scenario.\\

2. Strong empirical results.

The result of experiments shows great improvement on the selected base model (14.5% -> 36.5%), even comparable with larger model (e.g., DeepSeek-V3-0324) and commercial model (Claude Sonnet 3.5, gpt-4.1-2025-04-14) as reported in this paper and external reference (https://swe-rebench.com/).

**Weaknesses:**

1. Limited novelty in algorithmic design.

Both RFT and DAPO are not new. Previous work, such as Yuan et al., (https://arxiv.org/abs/2308.01825) has leveraged this technique to improve reasoning ability, and DAPO is from Yu et al., as cited. Some modifications, such as turn penalty, are more engineering rather than a theoretical innovation

2. Limited ablation to each component.

Although in Table.1, we can see improving performance as adding more stages, the effect of each component in the training pipeline is not fully discovered. For example, what if the RFT phrase or stage 1 in RL phrase is removed? By adding these comparisons, the necessity of each component will be more convincing.

3. Limited comparisons in perspective of RL algorithm and base model.

On one hand, In 3.2, we see a clear introduction of DAPO, but what makes DAPO better than GRPO or PPO  in SWE tasks remains unclear. Plus, how effective the modification to length penalty in DAPO is expected to be seen. Therefore I suggest an algorithmic analysis or experiment added in this paper.

On the other hand, though we see the improvements on Qwen2.5-72B-Instruct, two questions are naturally raised, why choose this model (any reasons in architecture or size?), and, what if we apply the proposed pipeline to another model?

4. Ambiguous connections among challenges, the proposed method, and findings.

Although in line 94, the proposed pipeline is claimed to address aforementioned challenges, I cannot directly link to how this pipeline becomes a solution, especially in the problem of sparse rewards. Further, in the second paragraph of 5.2, the authors attribute the degrading performance to distribution mismatch and claims performance will recover if sampling is unbiased. Please justify this issue deeper  and point out how the proposed training pipeline recovers the biased gradient updates.

**Questions:**

## Questions
1. The spec. of Qwen2.5-72B-Instruct shows the maximum of context length is 128k, but many results are tested on data with length up to 131k. Did you do any modification like truncation? Plus, you mentioned context parallelism in the appendix. Does this technique help deal with data with length exceed 128k?

2. In 4.2, the RFT yields 6548 trajectories. Do these come from 7249(tasks) * 10(runs)? and if so, how many tasks remain in this stage?

3. In line 314, you mentioned thousands of problems, are these data same as RFT's, filtered data (7249 tasks), or others?

4. In table 2, line 418, SWESynInfer-72B used the same base model as yours, but their "before" score is much higher than 11.4%. I founded they did maybe extensive prompt engineering, so I am wondering what if the propsed pipeline starts from this setting?

## Suggestion
Beside the suggestions appended on weaknesses,
1. More models with similar size can be compared as well. In Table.1, we see many rival model with various size other than 72B, and in Table 2, most models have 32B parameters. Therefore, a 32B model is suggested toward a fairer comparison.

------
If most of the weakness and questions are solved, I would be willing to increase my ratings.

---

> ### Author Response · Authors · 2025-11-20
> **Authors response**
>
> We thank the reviewer for their thoughtful feedback and for recognizing the timeliness of our topic and the strength of our empirical results (11.4% -> 39.0%). We address the reviewer's points below:
>
> 1: We acknowledge that the algorithms themselves are well-known. However, our contribution deliberately focuses on the unanswered question: "How can modern algorithms be effectively applied to complex stochastic environments where powerful LLMs operate within contexts of hundreds of thousands of tokens?". This specific application remains underexplored in the literature.
>
> 2: The RFT stage is performed primarily to accelerate training, with the main goal of improving instruction following and tool calling. We hypothesize that RL stages alone could achieve similar eventual performance but would require significantly more time. While we acknowledge that ablating this would strengthen the paper, we prioritized resources on the following ablations: (1) analyzing performance differences between biased and unbiased sampling during training, and (2) evaluating checkpoints in RL Stage 1 and 2 at both 65k context with 40 turns and 131k context with 80 turns to isolate the impact of Stage 2 from the simple increase in context length. The results for full SWE-bench Verified averaged across 10 runs are as follows:
>
> Stage 1 evaluated @ 65k 40turns: 35.7 ± 0.3%, pass@10: 54.6%
>
> Stage 1 evaluated @ 131k 80turns: 35.4 ± 0.3%, pass@10: 54.0%
>
> Stage 2 evaluated @ 65k 40turns: 37.9 ± 0.3%, pass@10: 56.4%
>
> Stage 2 evaluated @ 131k 80turns: 39.0 ± 0.5%, pass@10: 58.4%
>
> All new ablations will be included in the final version of the paper.
>
> 3: We selected DAPO as it was the state-of-the-art algorithm at the time of our experiments. As DAPO addresses many common RL training pitfalls, we built upon its findings, considering a re-evaluation of fundamental RL algorithms out of scope for this paper. Regarding model choice, we chose Qwen2.5-72B-Instruct as a representative high-capability open-weight model. While comparing multiple model families would be robust, the resource intensity of RL at this scale constrained us to deep ablations on a single strong backbone.
>
> 4: We will clarify the Line 94 formulation. Regarding Section 5.2, our new ablation shows that biased sampling caused validation performance to decrease from 35% to 18.8% after just 28 steps. This will be included in the final paper.
>
> **Questions:**
>
> 1: The official context support of Qwen2.5-72B-Instruct is 131,072 (2**17) tokens, as specified in config.json and on Hugging Face:  `Context Length: Full 131,072 tokens and generation 8192 tokens`. While often referred to as "128k" for simplicity, it factually supports 131k. Context parallelism is indeed essential for training in these scenarios: by splitting the sequence into chunks, it makes computing activations more feasible. Without it, Out-Of-Memory errors frequently occur even with contexts shorter than 131k.
>
> 2: That is correct. After filtering for trajectories that pass the test suite, 2121 unique tasks remain.
>
> 3: This refers to the same initial pool of 7,249 filtered instances.
>
> 4: That is correct, SWESynInfer-72B starts at 25.4%. This is because the authors use a more rigid scaffold with a fixed workflow (repository understanding -> fault localization -> patch generation). We deliberately chose a looser scaffolding where the agent operates freely, though we believe our conclusions regarding RL applicability hold regardless of the scaffolding choice.
>
> **Suggestions:**
>
> 1: We agree that exploring different model families and sizes would be valuable to further assess generalizability. However, given the significant computational resources required for long-context RL training and the strict time constraints, we prioritized the critical ablations (sampling bias and context scaling) described above. These were chosen to directly verify the soundness of our main claims within the available rebuttal window.

---

### Official Review · Reviewer_JZho · 2025-10-31

**Soundness:** 3
**Presentation:** 2
**Contribution:** 3
**Rating:** 6
**Confidence:** 3

**Summary:**

This paper addresses the challenge of applying reinforcement learning (RL) to large language models (LLMs) for multi-turn interactive tasks with stateful environments, a setting more aligned with real-world domains such as software engineering (SWE). The authors propose a methodology combining rejection fine-tuning (RFT) with execution feedback and a synchronous RL pipeline utilizing DAPO for iterative improvement. Using Qwen2.5-72B-Instruct as the base model, the pipeline achieves significant performance improvements on SWE-bench Verified and SWE-rebench benchmarks, showcasing the potential of the approach for training capable agents for multi-turn tasks.

**Strengths:**

1. The use of RL to train agents for multi-turn, stateful interactions is highly relevant for advancing LLM-based applications in real-world domains such as SWE. The focus on a structured RL pipeline for this problem is valuable and timely.
2. The combination of RFT and DAPO appears to be effective, as evidenced by the substantial improvements in Pass@1 scores on SWE-bench Verified and SWE-rebench. These results demonstrate the practicality of the approach, particularly when using open-weight models like Qwen2.5-72B-Instruct.
3. The experiments are built on a strong base model (Qwen2.5-72B-Instruct) and include comparisons with competitive benchmarks (e.g., DeepSeek-V3-0324, Qwen3-235B-A22B). The results are well-documented and provide convincing evidence of the proposed method's effectiveness.
4. The use of DAPO, particularly in the context of RL for SWE tasks, appears well-motivated and thoughtfully integrated into the pipeline.

**Weaknesses:**

1. While the results are promising, the novelty of DAPO as a contribution is not entirely clear. The paper would benefit from a clearer comparison with concurrent or prior approaches to clarify how DAPO differs from related methods. This is particularly important because RL for LLMs is a rapidly evolving field, and comparisons with recent work may be necessary to establish the significance of the contribution.
2. The paper's presentation could be improved. For example: The novelty of the work is not emphasized strongly enough in the introduction or methodology sections. It would help to explicitly highlight what aspects of RFT + DAPO are novel and how they advance the state-of-the-art. Some information feels basic or redundant. For instance, Figure 1 occupies a large amount of space but does not provide substantial insights beyond what is described in the text. Consolidating or reformatting such figures could make room for more detailed explanations of the methodology or comparisons with related work.

**Questions:**

1. How does the method scale with even more complex environments?
2. What are the computational costs of the RFT + DAPO pipeline compared to baseline fine-tuning or RL methods?
3. Are there specific failure modes observed during training or evaluation?

---

> ### Author Response · Authors · 2025-11-20
> **Authors response**
>
> We thank the reviewer for recognizing the relevance of our work to real-world SWE domains and the effectiveness of our structured pipeline. We address the reviewer's points below:
>
> 1: The primary novelty of our paper is demonstrating how RL scales when applied to large models (>8B) within long contexts (>8k) in stochastic, multi-turn scenarios. Previous literature lacked empirical benchmarks in this regime. As shown in Table 2, most existing SWE agents rely on distillation (e.g., from Claude 3.5). Among the few direct RL works, SkyRL-Agent shows modest gains (18% -> 21%), and DeepSWE is concurrent work. We will clarify this distinction in the final manuscript.
>
> 2: We will utilize the additional space in the final version to include two key ablations: (1) an analysis of biased vs. unbiased sampling (showing a drop from 35% to 18.8% when biased); and (2) a comparison of Stage 1 and Stage 2 models evaluated at both 65k with 40 turns and 131k contexts with 80 turns, showing following results (SWE-bench Verified, runs are averaged across 10 runs):
>
> Stage 1 evaluated @ 65k 40turns: 35.7 ± 0.3%, pass@10: 54.6%
>
> Stage 1 evaluated @ 131k 80turns: 35.4 ± 0.3%, pass@10: 54.0%
>
> Stage 2 evaluated @ 65k 40turns: 37.9 ± 0.3%, pass@10: 56.4%
>
> Stage 2 evaluated @ 131k 80turns: 39.0 ± 0.5%, pass@10: 58.4%
>
> This isolates the contribution of the two-stage training pipeline from the simple extension of the context window.
>
> **Questions:**
>
> 1: In Stage 2, we filter out tasks with high success rates, effectively shifting the training distribution toward more complex environments. The fact that our results continue to improve under this harder distribution demonstrates the method's ability to scale to higher difficulty levels.
>
> 2: In terms of data volume, the RL pipeline processed a total of ~9B tokens, whereas the RFT stage processed only ~0.2B tokens. It is also important to note that the RL stage is computationally heavier due to the inference cost of generating new rollouts. However, when compared to other RL baselines, DAPO is significantly more memory-efficient than PPO. While similar to GRPO, DAPO avoids the need for a separate critic model (or value head) required for GAE estimation, which is a critical factor when training 72B models on long contexts.
>
> 3: The primary failure mode we observed during training was instability caused by biased sampling. We analyzed this in our ablation: a run using biased sampling degraded from 35% to 18.8% Pass@1 within just 28 training steps. We will include a plot illustrating this performance collapse in the final paper.

---

### Official Review · Reviewer_ox7M · 2025-11-01

**Soundness:** 2
**Presentation:** 3
**Contribution:** 2
**Rating:** 2
**Confidence:** 3

**Summary:**

The paper trains a long-context, multi-turn SWE agent with a two-phase recipe: (1) Rejection fine-tuning (RFT) on self-generated, test-passing trajectories to fix tool-use/formatting issues; (2) synchronous on-policy RL with DAPO wih turn-level length penalty. Starting from Qwen2.5-72B-Instruct, they report Pass@1 on swe-bench verified improvement (from 11% to 39%) and swe-bench May/June respectively.

**Strengths:**

- This paper presents a clear, end-to-end agent training for multi-turn SWE and clear problem formulation with POMDP. Also, the presented two-phase recipe is standard yet effective, RFT follows by RL.
- I like the transparency of the good engineering and the negative results about the decoding mismatch part. The paper also cautions against decarding over-long trajectories (could hide looping and make the model not-generalizable). This provides good valuable guidance to the commnuity and rarely documented.

**Weaknesses:**

- The core ingredient: Rejection Sampling Finetuning (RFT) followed by on-policy RL is a well-executed application to various tasks with verifiable reward such as math reasoning task and single-turn/multi-turn code generation task. The algorithm design choice DAPO/GRPO is known and the new bit is tailoring the reward shaping to turn count and getting a big model to 131k context length stably. While the application and scaling are novel and successful, I’m missing a sharper “what’s truly new vs. engineered best-practice” positioning.
- There's very limited ablations study to justify the algorithm choice or the pipeline, for example ablating the proposed length penalty and varying different group size G. The point I mentioned in the Strength, though interesting as a read, lacks concrete number and experiments for backing:
  - How the training looks like for the decoding mismatch?
  - What's the perf when you decard over-long trajectories compared to the main exp. in the paper? Empirical evidence of model not being able to generalize to such looping trajectories in test time and maybe provide some stats and analysis?
  - The main selling point is the long-context training, could the author also provides test time performance when under different seq length to justify the effectiveness?

Overall, I read this paper as closer to an interesting tech report and a lot of claims are floating around without backing numbers and experiments. That being said, more empirical evidence are needed to justify the claims in the paper. This my main motivation of not recommending an acceptance.

**Questions:**

Please see above.

---

> ### Author Response · Authors · 2025-11-20
> **Authors response**
>
> We thank the reviewer for their thoughtful feedback and for appreciating the transparency of our engineering insights. We have used the rebuttal period to generate the requested empirical backing to address the "floating claims" concern.
>
> 1: While the RFT + RL recipe is established for short-horizon tasks, our primary contribution is solving the stability and efficiency challenges of scaling this to 131k-token contexts with a 72B-parameter model in a multi-turn stochastic environment. Prior works typically operate within 4-16k contexts on single-turn tasks. Vanilla RL applications in this regime often yield negligible gains (e.g., SkyRL: 18% -> 21.6% with Qwen3-14B, [1]). In contrast, our work provides the first empirical validation (alongside the concurrent DeepSWE work, [2]) of a stable recipe that drives a 3.5x improvement (11% -> 39%).
>
> 2.a: The reviewer asked for evidence regarding the impact of decoding mismatch. We conducted an ablation comparing our standard unbiased training run against a run with biased sampling (enabling standard decoding heuristics like top_p/top_k during rollouts). The run with biased sampling suffered immediate instability. Validation performance collapsed from 35% to 18.8% within just 28 training steps. We will include the training curve demonstrating this collapse in the final paper.
>
> 2.b: Regarding discarding over-long trajectories: our analysis shows that the model trained with length filtering failed to learn loop-breaking behaviors. This degraded model exhibited ~2x more exit_max_iter statuses (failing by exceeding the turn limit) during evaluation on Verified-50 compared to our main model.
>
> 2.c: To isolate the impact of our long-context training from simple inference-time context extension, we evaluated checkpoints from both RL Stage 1 and Stage 2 under both 65k with 40 turns and 131k with 80 turns settings (runs are on SWE-bench Verified and averaged across 10 independent launches):
>
> Stage 1 evaluated @ 65k 40turns: 35.7 ± 0.3%, pass@10: 54.6%
>
> Stage 1 evaluated @ 131k 80turns: 35.4 ± 0.3%, pass@10: 54.0%
>
> Stage 2 evaluated @ 65k 40turns: 37.9 ± 0.3%, pass@10: 56.4%
>
> Stage 2 evaluated @ 131k 80turns: 39.0 ± 0.5%, pass@10: 58.4%
>
> Simply extending the context for the Stage 1 model yields no gain. However, the Stage 2 model, explicitly trained with our long-context curriculum, effectively leverages the 131k window.
>
> All new ablation data will be included in the final version.
>
>
> [1] Shiyi Cao et al. SkyRL-v0: Train Real-World Long-Horizon Agents via Reinforcement Learning
>
> [2] Michael Luo et al. DeepSWE: Training a State-of-the-Art Coding Agent from Scratch by Scaling RL

---

### Author Response · Authors · 2025-12-02
**Author Comment to Area Chair**

Dear Area Chair,

Since the original reviewers are currently unable to update their scores to reflect our new experiments, we provide this summary to assist your assessment.

Reviewers reached a consensus on the timeliness and relevance of our work but identified a primary weakness: a lack of ablations to justify algorithmic choices.

During the rebuttal, we conducted significant new experiments to directly address this. We summarize the key quantitative findings below:

1. Justification for sampling strategy (addressing "decoding mismatch")
Reviewers questioned the necessity of our unbiased sampling approach versus standard decoding heuristics.

We compared our standard training against a run with biased sampling (enabling top-p/top-k during rollouts). The biased run collapsed immediately. Validation performance dropped from 35% to 18.8% within just 28 training steps.

2. Isolating RL gains from context extension

Reviewers asked if the final performance came from the RL training or simply the extended context window of 131k tokens. We evaluated checkpoints on SWE-bench Verified from two distinct RL training stages to isolate these factors:

Stage 1 (RL trained with 65k context):
- Evaluated @ 65k (40 turns): 35.7 ± 0.3% (pass@10: 54.6%)
- Evaluated @ 131k (80 turns): 35.4 ± 0.3% (pass@10: 54.0%)

Stage 2 (RL trained with 131k context):
- Evaluated @ 65k (40 turns): 37.9 ± 0.3% (pass@10: 56.4%)
- Evaluated @ 131k (80 turns): 39.0 ± 0.5% (pass@10: 58.4%)

These results support our conclusion that Stage 2 RL training with extended context indeed increases the agent's performance.

We believe these additional experiments provide the quantitative validation requested by the reviewers and directly resolve the questions regarding our design choices. We sincerely appreciate your time and consideration.

---

### Meta-Review · Area_Chair_hpSf · 2026-01-13

**Summary:**

This paper focuses on how to train long-context, multi-turn software engineering agents with reinforcement learning, and proposes a new method built upon RFT and DAPO. Experimental results based on Qwen-2.5-72B show that the proposed approach achieves substantial improvements in accuracy. However, the reviewers raised concerns regarding the research contributions and the experimental setup.

**Reviewer Concerns:**

1. The core contribution of the paper is not sufficiently clear. Both RFT and DAPO are existing methods, and the paper mainly emphasizes engineering and system-level innovations, while the technical contributions are not adequately articulated.

2. The experimental validation is insufficient, especially in ablation studies. All reviewers agreed that extra ablation experiments are necessary to demonstrate the importance and contribution of each component, including the impact of removing the RET stage, the first-stage RL training, the necessity of the decoding strategy, etc.

3. Method comparison and generalization analysis are inadequate. The experiments are conducted only on Qwen-2.5 as the base model, without comparisons across other backbone models. In addition, the training data is primarily based on SWE-bench, which raises concerns about potential overfitting.

**Reviewer Scores:**

The authors addressed some of the above concerns in their rebuttal by providing experimental results comparing different checkpoints from the first and second stages of RL. However, many of the ablation studies requested by the reviewers were still not covered, and the concerns regarding component-wise contributions remain largely unaddressed. In addition, the paper does not present a compelling and well-articulated statement of its research contributions, and it still reads more like a technical report than a mature research paper. Therefore, I think this paper is not yet ready for publication.

---

### Decision · Program_Chairs · 2026-01-26

Reject